# Genome sequence reconstruction using gated graph convolutional network

## Abstract

A quest to determine the human DNA sequence from telomere to telomere started three decades ago and was finally finished in 2021. This accomplishment was a result of a tremendous effort of numerous experts with an abundance of data, various tools, and often included manual inspection during genome reconstruction. Therefore, such method could hardly be used as a general approach to assembling genomes, especially when the assembly speed is important. Motivated by this achievement and aspiring to make it more accessible, we investigate a previously untaken path of applying geometric deep learning to the central part of the genome assembly—untangling a large assembly graph from which a genomic sequence needs to be reconstructed. A graph convolutional network is trained on a dataset generated from human genomic data to reconstruct the genome by finding a path through the assembly graph. We show that our model can compute scores from the lengths of the overlaps between the sequences and the graph topology which, when traversed with a greedy search algorithm, outperforms the greedy search over the overlap lengths only. Moreover, our method reconstructs the correct path through the graph in the fraction of time required for the state-of-the-art de novo assemblers. This favourable result paves the way for the development of powerful graph machine learning algorithms that can solve the de novo genome assembly problem much quicker and possibly more accurately than human handcrafted techniques.

## 1 Introduction

Fast and accurate *de novo* genome assembly is one of the most difficult problems in bioinformatics and it remains unsolved to this day. It focuses on reconstructing the original genomic sequence from a sample of shorter overlapping fragments, called reads, without any prior knowledge about the original sequence. The first major achievement of *de novo* genome assembly happened in the early 2000s when the Human Genome Project was finished, an effort that took over a decade and cost billions of dollars (Lander et al., 2001). The reported results of the project were that 99% of the genome had been reconstructed with less than 400 gaps. Unfortunately, that was not entirely correct, as only the euchromatic portion of the genome was considered while the heterochromatin was left out. When the heterochromatin regions—which include centromeres, telomeres, and tandem gene arrays—are also taken into account, the final result is that more than 5% of the entire genome was either missing or incorrect.

Since then, the sequencing technologies have improved significantly in many ways, but most notably in terms of the lengths and accuracies of the reads they produce. At the forefront of the latest sequencing technologies are the HiFi reads developed by PacBio (Wenger et al., 2019) and the ultra-long reads by Oxford Nanopore Technologies (Jain et al., 2018), both of which were crucial for the most recent breakthrough in the field of de novo genome assembly—full reconstruction of the entire human genome, with no regions left unsolved (Nurk et al., 2021). Nevertheless, this was enabled not only by the latest sequencing technologies, but also by a tremendous effort of numerous researchers and bioinformaticians who used various *de novo* assembly tools and manually inspected large genomic regions.

One of the more common approaches to de novo genome assembly, which was also the one used in the recent reconstruction of the human genome, is the Overlap-Layout-Consensus (OLC) paradigm.

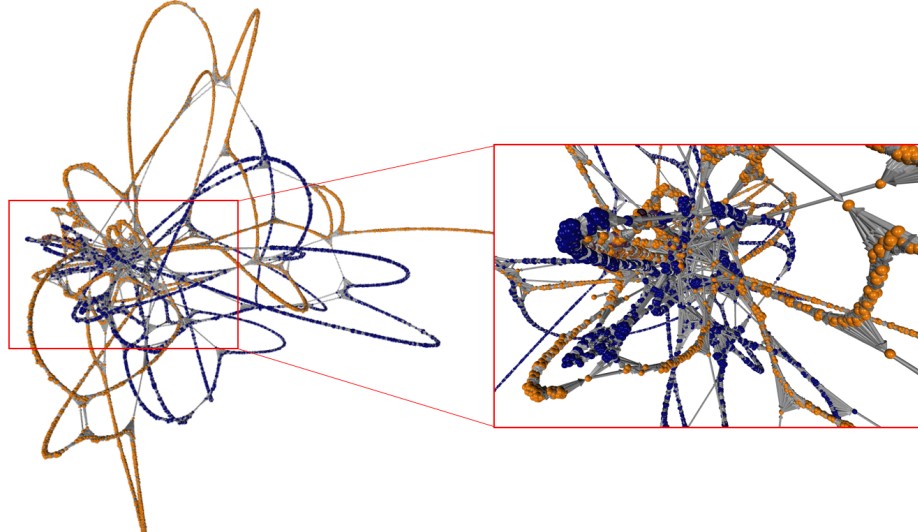

Figure 1: An assembly graph of one part of chromosome 11, containing the repetitive region. A zoomed-in view of the repetitive region can be seen on the right. In the graph, orange nodes represent nodes taken for the optimal genome reconstruction, while the blue nodes should be avoided. Figure generated with Graphia (Freeman et al., 2020).

In the Overlap phase, the reads in the sample are mapped onto each other in an all-versus-all manner in order to find overlaps between them. All reads that are entirely contained in other reads are removed from further processing. From the rest of the overlapped reads, an assembly graph is built—a directed graph in which nodes represent reads and edges represent the suffix-prefix overlaps between the reads. In the Layout phase, the assembly graph is simplified in order to find a path through it that would reconstruct the original genome. Finally, in the Consensus phase, all the reads are aligned to the reference in order to clean the assembly sequence of errors that happened during the rest of the process.

In an ideal scenario, the Layout phase would be formulated as finding a Hamiltonian path over the assembly graph—visit every node in the graph exactly once. However, due to imperfect read qualities, sequencing artifacts, and repetitive genomic regions, finding a Hamiltonian path is usually not possible and an unknown number of nodes and edges has to be removed. Because of this, instead of finding a path through the graph directly, modern assemblers rely on heuristics to simplify the entire graph, by iteratively removing nodes and edges deemed unnecessary, such as removing transitive edges, trimming tips, and popping bubbles (Li, 2016; Vaser & Šikić, 2021). Frequently, however, the assembly graphs are highly complex, and even this is not a straightforward task, as some regions cannot be simplified by the current heuristics. One such region can be seen in Figure 1 on the right. For lack of a better method, these complex regions are cut out of the graph, and we are left with multiple fragments of the original genome instead of the continuous sequence. To this day, the problem of fragmentation continues to plague all the existing assemblers.

The central part of reconstructing the full human genome was untangling these complex regions in the assembly graph instead of cutting them. This approach required an abundance of data, various tools, and a detailed manual inspection of certain regions and individual reads. Reproducing such efforts on new genomes seems infeasible, and thus fast and accurate de novo genome assembly remains elusive.

In this work, we propose a novel approach to de novo genome assembly, one based on deep learning and finding a path through the graph instead of relying on handcrafted heuristics to simplify the graph. We train a non-autoregressive model based on gated graph convolutional network (GatedGCN) introduced by Bresson & Laurent (2017) that takes an assembly graph and outputs a score for each edge. These scores can then be used to guide a search algorithm over the graph, producing a path that represents the reconstructed genome. Our idea is illustrated in Figure 1, where the orange nodes represent the nodes traversed in the optimal path, and blue the ones that should be avoided. For

this, we create a synthetic dataset of reads from the real human genomic data, and generate assembly graphs prior to any simplification steps in order to avoid any errors such simplifications might produce. We believe, this approach could reduce the fragmentation of the reconstructed genomes as well as the runtime of the assembly process, since some simplification steps can be time-consuming. Thus, we make both the codes[1] and the data[2] available for other researchers to use and join in on tackling the *de novo* genome assembly using deep learning.

Lately, graph neural networks have been applied to a variety of biological problems, ranging from drug design (Stokes et al., 2020) and protein interactions (Gainza et al., 2020), to predicting anti-cancer foods (Gonzalez et al., 2021). However, to the best of our knowledge, the only work based on geometric deep learning that addresses de novo genome assembly is Vrček et al. (2020), which is more of a proof-of-concept work that simulates the deterministic simplification algorithms in the Layout phase, instead of tackling the tangled regions themselves. Moreover, the data used in Vrček et al. (2020) was completely synthetic, not generated from the biological sequences. Here, we go a step further by starting with the complete human genome and focusing on one of the main problems in genome assembly.

The rest of the paper is composed as follows: In Section 2 we describe the dataset preparation, in Section 3 we formulate the problem mathematically, in Section 4 we describe the architecture of the used model, in Section 5 we state the performed experiments, and in Section 6 we discuss the results. Finally, Section 7 concludes this paper.

## 2 DATASET

Generating the dataset can roughly be separated into three tasks. First, we simulate the genomic data from the human genome. Second, we adapt an existing tool for *de novo* genome assembly, Raven (Vaser & Šikić, 2021), to build an assembly graph for each set of reads. Finally, we implement an algorithm that leverages the information stored during the read-simulation phase and finds a ground-truth path through an assembly graph. We use these paths as the supervision signal during the training phase.

### 2.1 SIMULATING READS

To simulate the reads faithfully, we start with the recently reconstructed human genome, called CHM13 (Nurk et al., 2021), which consists of one chromosome from each of 23 pairs of human chromosomes. In the assembly process, each chromosome would ideally be represented by a single component of the large disconnected graph of the entire human genome, but in a realistic case, several chromosomes could be connected into a single component. Starting from a simpler scenario, we isolated one chromosome and split it into shorter "mini-references", each 2 million base pairs (bp) long.

During the sequencing process of the real samples, all information about the ordering of the reads and their position on the genome is lost. This is the main reason why we are not using sequenced human data, but simulate our own by sampling the reads from the mini-references. Thus, we are able to store the positional information for each read, which allows us to construct the ground-truth path for training.

The reads are simulated by mimicking the sequencing process. The mini-reference is sampled in a manner that each base of the mini-reference is covered approximately 20 times, which represents the number of times genomes are copied and fragmented prior to sequencing. The lengths of the reads are sampled from a normal distribution with the mean value of 20,000 and the standard deviation of 1,500. These values were determined empirically so that the simulated reads would resemble the real PacBio HiFi reads. In contrast to sequencers, we introduce no errors to the simulated data to facilitate. Considering that the average accuracies of the real PacBio reads are usually over 99.5% and that there are tools that perform error correction of reads (Cheng et al., 2021), the assumption of errorless reads is not too far-fetched.

---

[1]Code: `https://anonymous.4open.science/r/gnn-genome-reconstruction-4462`
[2]Data: currently available under supplementary materials

## 2.2 Generating graphs

Once the reads are simulated, we can construct the graphs. For this purpose, we use an adapted version of Raven, an assembler that follows the OLC paradigm and can output the assembly graphs at different stages of the Layout phase (Vaser & Šikić, 2021). Additionally, we required Raven to keep only the perfect overlaps between the reads. Therefore, unless two reads have a suffix and prefix which are matching in all the bases, they will not be connected with an edge in the assembly graph.

At the end of the Overlap phase, the assembly graph is constructed. We output the generated graph at the start of the Layout phase, prior to any simplification algorithm applied, in order to avoid errors that can occur during the simplification steps. The end result of this entire process is 50 graphs, each containing around 3,500 nodes and 50,000 edges. Considering that the reads are perfect and the similarity score of each overlap is 1.0, we only rely on overlap lengths as edge features, and use no node features (or rather, we specify feature of each node to be 1). A possible approach would be to encode the genomic sequences with a 1D-CNN and use the read-sequence encodings as node features and overlap-sequence encodings as additional edge features, but we leave that for future work. Therefore, in order to train on the generated graphs, the last thing needed is the supervision signal—a ground-truth path for each graph.

## 2.3 Ground-truth paths

A ground-truth path is a path in the assembly graph that produces the longest length of the reconstructed genome. For this, we utilize the positional information that was stored for each read during the sampling process and implement an algorithm resembling depth-first search, with an additional preference for successor nodes that are closer to the current node on the mini-reference. The neighbors that don't share position on the mini-reference with the current node are avoided. Although the reads and the overlaps in the created graphs are both perfect, connections between the distant genomic regions can still exist. The main culprits for this are repetitive regions, regularly found in telomeres, centromeres, and highly duplicated ribosomal RNA genes.

In a way, this algorithm can be described as an exhaustive search with an oracle. Even though the oracle—the reads' positions on the mini-reference—guarantees that the reconstructed genome will be optimal, we noticed cases when the length of the reconstruction was less than of the original mini-reference. There are two reasons for this. First, during the Overlap phase, all the reads are trimmed, which slightly reduces them in length. Since the ends of the mini-reference are covered only by the ends of the reads, trimming the reads necessarily leads to loss of information. Second, and the more interesting case, happens due to an error during the Overlap phase resulting in fragmented assembly graph. This occurs even though the sampled reads can cover the entire mini-reference (apart from a few bases on either end). We believe this happen due to a combination of repetitive regions and discarding of reads completely contained inside the other reads. However, more analysis is needed to make a definite claim.

## 3 Formulating the problem

Let $\mathcal{G}(V, E)$ be the graph obtained from a set of reads $R$. A node $i \in V$, represents a read $r_i \in R$, while an edge $i \rightarrow j \in E$ represents a suffix-prefix overlap between reads $r_i$ and $r_j$. For a node $i$ in the graph $\mathcal{G}$, there exists a node feature $x_i \in \mathbb{R}^{d_v}$, while for an edge $i \rightarrow j$ exists an edge feature $z_{ij} \in R^{d_e}$, where $d_v$ and $d_e$ are the dimensionalities of the node and edge features, respectively.

Given such a graph, our objective is to identify the sequence of nodes (i.e., a path), which reconstructs the genome optimally:

$$(i_1^*, \ldots, i_n^*) = \arg\max_{(i_1, \ldots, i_n)} \text{RecLen}(i_1, \ldots, i_n), \tag{1}$$

where RecLen is the length of the reconstructed genome for the given node sequence. Finding such a path in an exact manner would be an NP-hard problem, so we need to reformulate our objective in a probabilistic way: given graph $\mathcal{G}(V, E)$, identify a sequence of nodes which maximizes the conditional probability that the chosen sequence is optimal:

$$P(i_1, \ldots, i_n \text{ is optimal } \mid \mathcal{G}) = \tag{2}$$

$$= P(i_1 = i_1^* \mid \mathcal{G}) \cdot P(i_2 = i_2^* \mid \mathcal{G}, i_1 = i_1^*) \cdot \ldots \cdot P(i_n = i_n^* \mid \mathcal{G}, \ldots i_{n-1} = i_{n-1}^*) \tag{3}$$

$$\approx \prod_{k=1}^{n-1} P(i_{k+1} = i_{k+1}^* \mid \mathcal{G}, i_k = i_k^*) \approx \prod_{k=1}^{n-1} p_{i_k, i_{k-1}}. \tag{4}$$

In the last line, we introduce two approximations. First, we assume that the conditional probability of choosing the correct next node depends only on the current node and not on all the previous choices. Second, the exact conditional probabilities are unknown, and thus we approximate them with scores $p_{ij} \approx P(i_{k+1}^* = j \mid \mathcal{G}, i_k^* = i)$. Therefore, our task comes down to finding a way to compute the scores $p_{ij}$ so that they would resemble the real conditional probabilities as much as possible. We propose two approaches:

**Naive approach.** Considering that in the current setting we use no node features and use only overlap length as the edge features, a naive approach would be to normalize overlap lengths over the neighborhood:

$$p_{ij} = \frac{\text{len}(i \to j)}{\sum_{i \to j'} \text{len}(i \to j')}, \tag{5}$$

where, for sake of simpler notation, $\text{len}(i \to j)$ denotes the length of the overlap between reads $r_i$ and $r_j$. Longer overlap between two reads indicates that they might be closer on the reference and thus the lesser chance of making an error in the traversal.

**GNN approach.** The second approach is based on training a graph neural network to compute the scores. We deem that such a model, with enough data and expressive power, might be able to leverage the topology of the graph and thus produce scores of higher quality.

Once these scores are computed, a search algorithm guided by these scores can be run on the graph. In this work, we consider greedy search which will always choose the highest score. This implies that, in case the scores don't faithfully represent the conditional probabilities, the chosen sequence of nodes will certainly be suboptimal. Different choices of search algorithms are also possible, e.g., beam search, but we leave that for future work. Notice that, in a setting where scores are predicted with the naive approach, and the search algorithm is greedy, the entire task comes down to greedily choosing edges with the longest overlap.

## 4 MODEL ARCHITECTURE

The proposed model for obtaining the probability scores is non-autoregressive and can be split into three parts—encoder, processor, and decoder.

**Encoder.** A layer that transforms node features $x_i \in \mathbb{R}^{d_v}$ and edge features $z_{ij} \in \mathbb{R}^{d_e}$ into the $d$-dimensional node and edge representations. As stated in Section 2, we use $x_i = 1.0$ and $z_{ij} = \text{len}(i \to j)$, thus making $d_v = d_e = 1$. Encoder for both node and edge features is a single linear layer:

$$h_i^0 = W_1 x_i + b_1 \in \mathbb{R}^d, \tag{6}$$

$$e_{ij}^0 = W_2 z_{ij} + b_2 \in \mathbb{R}^d, \tag{7}$$

where $h_i^0$ is the initial node representation of the node $i$, $e_{ij}^0$ is the initial representation of the edge $i \to j$, and $W_1 \in \mathbb{R}^{d \times d_v}, W_2 \in \mathbb{R}^{d \times d_e}, b_1, b_2 \in \mathbb{R}^d$ are learnable parameters.

**Processor.** The main part of the network consists of multiple GNN layers. For this task, we modify the GatedGCN (Bresson & Laurent, 2017) to perform on directed graphs by including the information from both the predecessors and successors of every node. In addition to the original GatedGCN, we include the edge feature representations, and use a dense attention map $\eta_{ij}$ for the edge gates, as proposed in Bresson & Laurent (2019); Joshi et al. (2019).

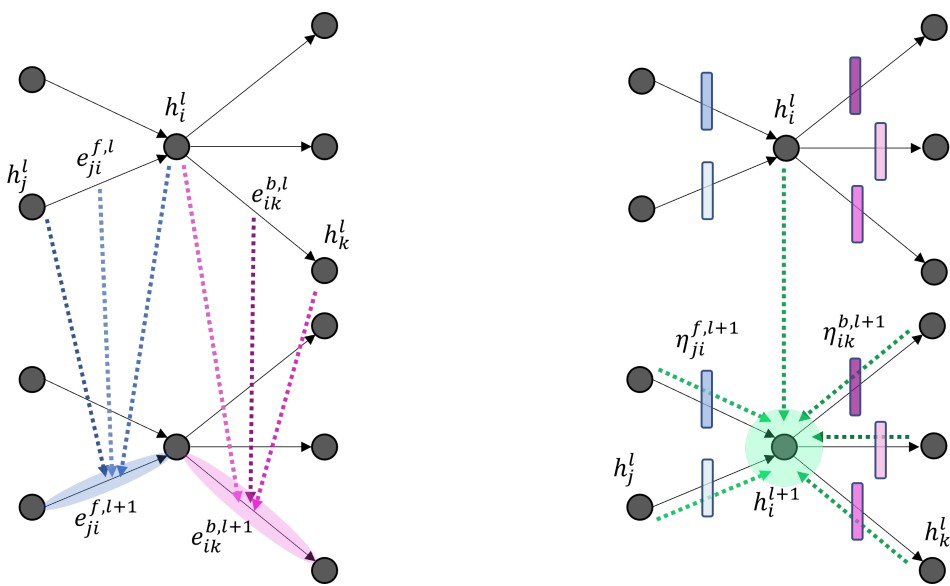

Figure 2: Left: updating the edge representations, both forward (blue) and backward (magenta). Right: Updating the node representation (green).

The motivation for using this layer comes mainly from its performance on the Traveling Salesman Problem (Joshi et al., 2019). Since the problem of finding the optimal walk on the assembly graph is similar to that of TSP, it makes sense to reuse the architecture. In addition, it was shown that the GatedGCN outperforms many other models in several tasks (Dwivedi et al., 2020), which gives us further reason to use this architecture.

Since we are working with the directed graphs, we will distinguish between the messages passed along the edges (forwards), and in the opposite direction of the edges (backwards). Thus, we will also use two sets of edge representations at a layer $l$, forward edge representations $e_{ij}^{f,l} \in \mathbb{R}^d$ and backward edge representations $e_{ij}^{b,l} \in \mathbb{R}^d$. For the initial values of both of them, we choose the initial edge representation obtained from the encoder, $e_{ij}^{f,0} = e_{ij}^{b,0} = e_{ij}^0$. Accordingly, we also use forward and backward edge gates at a layer $l$, $\eta_{ij}^{f,l}$ and $\eta_{ij}^{b,l}$ respectively.

Let now node $i$ be the node whose representation we want to update, and let its representation after a layer $l$ be $h_i^l$. Also, let all the predecessors of node $i$ be denoted with $j$ and all its successors with $k$. Then, in the layer $l+1$, the node and edge representations will be:

$$h_i^{l+1} = h_i^l + \text{ReLU}\left(\text{BN}\left(A_1^l h_i^l + \sum_{j \to i} \eta_{ji}^{f,l+1} \odot A_2^l h_j^l + \sum_{i \to k} \eta_{ik}^{b,l+1} \odot A_3^l h_k^l\right)\right) \in \mathbb{R}^d, \quad (8)$$

$$e_{ji}^{f,l+1} = e_{ji}^{f,l} + \text{ReLU}\left(\text{BN}\left(B_1^l e_{ji}^{f,l} + B_2^l h_j^l + B_3^l h_i^l\right)\right) \in \mathbb{R}^d, \quad (9)$$

$$e_{ik}^{b,l+1} = e_{ik}^{b,l} + \text{ReLU}\left(\text{BN}\left(C_1^l e_{ik}^{b,l} + C_2^l h_i^l + C_3^l h_k^l\right)\right) \in \mathbb{R}^d, \quad (10)$$

where all $A, B, C \in \mathbb{R}^{d \times d}$ are learnable parameters, ReLU stands for rectified linear unit, BN for batch normalization, and $\odot$ for Hadamard product. The edge gates are defined as:

$$\eta_{ji}^{f,l} = \frac{\sigma\left(e_{ji}^{f,l}\right)}{\sum_{j' \to i} \sigma\left(e_{j'i}^{f,l}\right) + \epsilon} \in \mathbb{R}_+^d, \qquad \eta_{ik}^{b,l} = \frac{\sigma\left(e_{ik}^{b,l}\right)}{\sum_{i \to k'} \sigma\left(e_{ik'}^{b,l}\right) + \epsilon} \in \mathbb{R}_+^d \quad (11)$$

where $\sigma$ is the sigmoid function, and $\epsilon$ is a small value in order to avoid division by zero. This rather complicated updating procedure is illustrated in Figure 2.

**Decoder.** A multi-layer perceptron (MLP) decodes the obtained representations into the probability scores. Probability score $p_{ik}$ for traversing an edge $i \rightarrow k$ is computed from node representations of nodes $i$ and $k$, as well as both forward and backward edge representations of the edge $i \rightarrow k$, all after the final GatedGCN layer $L$:

$$p_{ik} = \text{MLP}(h_i^L \parallel h_k^L \parallel e_{ik}^{f,L} \parallel e_{ik}^{b,L}), \tag{12}$$

where $(\cdot \parallel \cdot)$ is the concatenation operator.

## 5 EXPERIMENTS

### 5.1 TRAINING

Processing of a graph during one training epoch is done iteratively. In a single iteration, the graph is fed to the model providing us with scores for all the edges. We start the traversal from the starting node of the ground-truth path and take $w$ steps, where $w$ is a hyperparameter and we refer to it as the walk length. In each step, best successor is predicted from their probability scores, while the correct next node is obtained from the ground-truth path. The cross-entropy loss is computed over the current node's successors, followed by teacher forcing where we choose the successor given by the ground-truth as the next node. This is repeated for $w$, after which the copmuted losses are avereged, the backpropagation is performed, and the next iteration starts, continuing from the last visited node. This is repeated until the end of the ground-truth walk. It is important to notice that, while computing the loss, the number of candidates in each step can vary, as the number of successors also varies.

The training was performed on a dataset consisting of 50 graphs with a 30/10/10 train/validation/test split. Each graph had about 3,500 nodes and 50,000 edges, all of them generated from 2 Mbp mini-references coming from chromosome 11. We used Adam optimizer (Kingma & Ba, 2014), with the inital learning learning rate of $10^{-4}$. We also decay the learning rate multiplying by with 0.9 in case there was no improvement in the validation loss for 5 epochs. The evaluation metric used during training to keep track of the learning process was the accuracy of predicting the best next neighbor, which we also refer to as the local reconstruction metric. The entire training was done on a single Nvidia A100 GPU.

**Node-level prediction accuracy.** Predicting the best successor is the only metric used during the training, since it is easy to calculate the corresponding loss and accuracy. In a way, the better the model follows the ground-truth, the better it should reconstruct the entire sequence. However, there are some situations where choosing the incorrect node makes no difference (e.g., transitive edges such as $i \rightarrow k$, when there also exist $i \rightarrow j$ and $j \rightarrow k$), as well as situations where choosing the correct node is crucial (e.g., dead-end nodes). Therefore, even though used for training, this is not the best metric for genome assembly in general. Still we expect a certain transferability between the local and the global task, and this approach simplifies the training significantly. In addition, we also perform teacher forcing during the validation and testing—even though the model makes a mistake, we will put it back on the right path for the sake of easier evaluation of the training process.

### 5.2 INFERENCE

At inference, the predictions are performed not performed iteratively, but all in one go—we feed the graph to the model and find the path with a greedy algorithm, depending on the probability scores. The greedy algorithm runs until it reaches a node without outgoing edges or all of the node's successors have already been visited. Since there is no good option for choosing the starting node, we run the greedy search from all the nodes that have in-degree zero and choose the longest walk. At inference, we cannot evaluate the performance by the accuracy of choosing the best next neighbor, since there is no ground-truth. Moreover, genome assemblers usually work in a completely different way, so this would make the comparison against them impossible. Therefore, we evaluate our model on two other tasks—length of the reconstruction and the execution time.

**Reconstructed sequence length.** We measure the reconstructed length as the percentage of the original mini-reference length from which the graph was generated. This metric is robust to errors

Table 1: Comparison of GNN, naive approach, Raven, and exhaustive search.

| Method | 2 Mbp | | 5 Mbp | | 10 Mbp | |
|---|---|---|---|---|---|---|
| | length [%] | time [s] | length [%] | time [s] | length [%] | time [s] |
| **ES***  | $99.79 \pm 0.16$ | $2.5 \pm 0.1$ | $95.82 \pm 14.67$ | $7.5 \pm 1.1$ | $95.93 \pm 13.94$ | $23.0 \pm 5.0$ |
| **Greedy** | $93.50 \pm 17.88$ | $0.2 \pm 0.0$ | $90.21 \pm 24.01$ | $0.7 \pm 0.1$ | $82.11 \pm 33.14$ | $1.4 \pm 0.6$ |
| **Raven** | $90.85 \pm 26.74$ | $18.0 \pm 0.5$ | $93.68 \pm 17.25$ | $65.0 \pm 2.0$ | $95.92 \pm 13.96$ | $131.0 \pm 5.0$ |
| **GNN** | $99.20 \pm 1.86$ | $0.5 \pm 0.1$ | $93.51 \pm 18.31$ | $1.4 \pm 0.3$ | $95.73 \pm 13.90$ | $3.0 \pm 0.4$ |

while choosing transitive edges, but harshly punishes choosing dead-end nodes and wrong paths in general. Ideally, the length ratio would be 1, but it is expected that it will be slightly less due to the way the reads are simulated and processed prior to the Layout phase (e.g., trimming of reads mentioned in Section 2.3).

**Execution time.** One of the main pitfalls for many assemblers is their execution time. We evaluate the time needed for the model to process the graph and the search algorithm to find the best path through it. All the experiments related to execution time were performed on a single Intel Xeon E5-2698 v4 CPU.

**Benchmark.** We benchmark our model on the mentioned tasks against three other approaches. First, the naive approach where the scores are calculated as normalized overlap lengths, as proposed in 3. This approach comes down to running a greedy algorithm over the overlap lengths, and thus refer to it as the greedy in the next section. Second approach is the one used for obtaining the ground-truth paths as explained in Section 2.3, the exhaustive search with oracle. Finally, we also compare the developed model against an existing assembler Raven, which was used to generate the assembly graphs in Section 2.2.

## 6 RESULTS

The results reported here were obtained by the best-performing model, which had 8 GatedGCN layers, latent dimension 32, and the walk length was 10. The MLP classifier consisted of a single layer. During training this model achieved a $99.90\%$ accuracy on the test set and upon deeper inspection, we noticed that the errors mostly consisted of traversing transitive edges.

Table 1 shows the evaluation results of our model (GNN) benchmarked against greedy approach, Raven, and exhaustive search with oracle (ES*). For the reported lengths, first the percentage of the reconstruction was calculated for each graph, after which all the percentages were averaged. With the aim to test how our model scales to larger graphs, and thus larger genomes, we trained it on graphs generated from 2 Mbp mini-references, and evaluate on graphs generated from 2, 5, and 10 Mbp mini-references. The number of nodes and edges scaled linearly with the length of the mini-reference. For the evaluation on graphs generated from 2 Mbp mini-reference, we reuse the graphs from the test set, coming from chromosome 11. Additionally, we cut chromosome 10 into 5 Mbp mini-references and chromosome 12 into 10 Mbp mini-references. There was no particular reason for choosing the chromosomes 10, 11, and 12.

**GNN vs ES*.** The developed GNN-based method doesn't manage to reconstruct the genomes as well as the exhaustive search, but does not fall far behind either. The difference comes from the cases where our model got stuck in a dead-end node, similarly to what happened to the greedy algorithm.

**GNN vs Greedy.** We notice that our model consistently outperforms the greedy algorithm in terms of reconstructed length. This means that the model managed to leverage the graph topology in order to avoid some pitfalls where the greedy algorithm got stuck, such as dead-end nodes. The greedy algorithm is slightly faster than our model, which comes as no surprise. Interestingly, the difference

in execution times is relatively small, mainly because the neural network we used is also relatively small so a single forward pass through it can be efficiently done on a single CPU.

**GNN vs Raven.** We notice that Raven slightly outperforms the GNN model on the two larger datasets, but on the 2 Mbp dataset falls behind even the greedy approach. After a more thorough analysis, we noticed that Raven underperformed on only one graph, while on all the other graphs in that dataset it was as good as the GNN. The graph on which Raven failed came from a highly repetitive region, and since Raven tries to simplify the graph by removing nodes and edges, it ends up cutting the graph into numerous fragments, the longest one being only around 15% of the original mini-reference. GNN managed to correctly find a path through that graph. This is a critical result, as the repetitive regions are the main reason why the assemblers fail to accurately reconstruct genomes. At the same time, our method clearly outperforms Raven in terms of speed.

**Generalization.** We show that the model performs well consistently over graphs of different sizes, both in terms of accuracy and speed. Yet, human chromosomes are up to 250 Mbp long, so a definite conclusion on how it would perform in such a setting cannot be made.

## 7 CONCLUSION

In this work, we introduce a novel approach to solving *de novo* genome assembly based on graph neural networks and finding a path through the assembly graph. We created a dataset of assembly graphs based on real human genomic data on which the developed model was trained and evaluated. The model was also evaluated against a naive greedy approach, an exhaustive search using the positional information of reads, and an existing genome assembler Raven (Vaser & Šikić, 2021). It was shown that it consistently outperforms the greedy approach in terms of reconstructed length and Raven in terms of execution speed. More interestingly, it was shown that the model outperformed Raven when given a highly complex graph from a repetitive region.

These results are particularly promising to solve challenging regions more accurately and in far less time than the existing assemblers can. Future work will investigate the proposed technique on sequenced instead of generated data, with the ultimate aim of using it as a tool for untangling the assembly graphs to reduce the fragmentation and the execution time. We believe that combining path-finding techniques with deep learning will play a major role in improving *de novo* genome assembly.

### REPRODUCIBILITY STATEMENT

All the codes and the data used in this paper are made available, apart from the simulated reads due to their size, but the reads are not necessary for reproducing the results as we include the assembly graphs built from those reads. The human reference genome CHM13 which we used to generate the mini-references, as explained in Section 2, is publicly available here: `https://github.com/marbl/CHM13`. The codes used to simulate the reads from the mini-references are available together with the rest of the code here `https://anonymous.4open.science/r/gnn-genome-reconstruction-4462`. For generating assembly graphs we used an adapted version of Raven assembler, which is available here: `https://anonymous.4open.science/r/raven-D8F7`. In order to make the efforts to reproduce our results as easily as possible, we also make available the dataset we used for training and evaluation, as well as the best performing model—both available as supplementary material (this includes the assembly graphs and additional information). The instructions to help reproduce the results are available with the rest of the code.

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
