# OpenReview forum: "Genome Sequence Reconstruction Using Gated Graph Convolutional Network"
_ICLR.cc/2022/Conference — ICLR 2022 Submitted_

### Official Review · Reviewer_c7uL · 2021-11-01

**Correctness:** 4
**Technical Novelty And Significance:** 3
**Empirical Novelty And Significance:** 1
**Recommendation:** 3
**Confidence:** 4

**Main Review:**

The benchmarks aren’t sufficient to demonstrate the potential for this approach. The authors do not use standard metrics for this field or compare to widely used genome assemblers like wtdbg2 and mdBG. These manuscripts are also great references for additional metrics to study.

Ekim, B., Berger, B. & Chikhi, R. Minimizer-space de Bruijn graphs: Whole-genome assembly of long reads in minutes on a personal computer. Cell Syst 12, 958-968.e6 (2021).
Ruan, J. & Li, H. Fast and accurate long-read assembly with wtdbg2. Nat Methods 17, 155–158 (2020).

The use of only simulated data is another problem. There is often a very large difference between solving this problem on simulated versus real data. For this reason, the field has needed to hold competitions to properly benchmark methods, such as the Assemblathon.

The authors’ simulated data also contains no sequencing errors. I completely disagree with the authors that this assumption is not “far-fetched”. The authors must benchmark their method on data containing sequencing errors, which are an inevitable presence in real data. If they believe error correction methods are a useful stage in a full pipeline, they are welcome to include such a method.

On p. 7, the authors write, “We start the traversal from the starting node of the ground-truth path and take w steps, where w is a hyperparameter and we refer to it as the walk length." How do you make use of data from the later portion of the genome if every SGD step begins from the start and goes for 10 graph steps?

**Summary Of The Paper:**

The authors propose a graph neural network approach for genome assembly. In the framework of overlap-layout-consensus, a graph structure is formed in which reads are nodes and overlaps between them are edges. In theory, one desires a Hamiltonian path that visits every node and thus forms a reconstruction of the full genome.

The authors suggest that a gated graph convolutional neural network could compute on this graph to identify the true edges which should be followed in the Hamiltonian path. This is a creative and intriguing direction of research.

They specify a procedure to simulate data from a known genome and train the algorithm. They benchmark their method on simulated data versus another recently published genome assembler called Raven.

**Summary Of The Review:**

Processing assembly graphs with graph neural networks is an intriguing direction, but this work hasn't sufficiently demonstrated its value.

---

> ### Author Response · Authors · 2021-11-18
> **Response to Reviewer c7uL**
>
> We thank the reviewer for the insightful review.
>
> Due to some frequently received comments, we decided to address them as a general reply found here: https://openreview.net/forum?id=1QxveKM654&noteId=ZnkbPVBdr_7
>
>
> - Standard metrics
>
> We do not use standard metrics since most of them are irrelevant in our case of finding a path in a graph constructed from perfect reads. For example, the found path would be one (and only) contig, so the measures like N50 would not make a lot of sense at this point. However, we agree that eventually including these measures will be necessary once we construct a full assembler, but this was out of scope of this work.
>
>
> - mdBG and wtbg2
>
> mdBG and wtdbg2 use a different approach to build and solve the graph (de Bruijn graphs), and at this point we only focused on showing that GNNs could be used in the Layout phase for solving the assembly graphs. That’s also the reason why we benchmarked only against Raven, since we build the graph using Raven’s Overlap phase.
>
>
> - Simulated vs real data
>
> Yes, we cannot claim it is fully generalizable to real data, and the usage of real data was left for future work. However, we argue that once the graph is constructed the influence of read errors is neglectable. Yet, they have a high influence on the overlap phase. In addition, we have found even usage of 100% accurate reads does not guarantee a construction of a graph that could be traversed from end to end. The problems are influenced with overlap algorithms which underperform in repetitive regions mostly due to the intention to reduce running times.  Applied simplification helped us to improve the overlap phase in Raven and we argue that at this moment the usage of perfect reads is a reasonable first approximation.
>
>
> - Graph traversal
>
> Every SGD step does not begin from the start but from the walk_length * n, where n is the iteration index over that graph. For example, in case of walk_length = 10, the first iteration starts from 0 (in a zero-based indexed case) and finishes at 10 (10 predictions in total), the second starts at 10 and finishes at 20 (again 10 prediction in total), until the end. The last iteration could contain less than 10 steps, but the individual losses are averaged over the number of steps, so it doesn’t matter.

---

> > ### Comment · Reviewer_c7uL · 2021-11-30
> > **Maintain my score**
> >
> > Thank you to the authors for their response to my comments. I still believe the authors would need to do more to demonstrate their algorithm before it would make for a compelling publication in ICLR. Therefore, I maintain my score.

---

### Official Review · Reviewer_xJLL · 2021-11-01

**Correctness:** 2
**Technical Novelty And Significance:** 2
**Empirical Novelty And Significance:** 3
**Recommendation:** 3
**Confidence:** 4

**Main Review:**

Strengths:
+ The paper presents an interesting idea of using Graph-based deep learning to learn the path for genome assembly thus extending the previous neural network-based assembly idea to graphs.
+ The paper is clearly written. The problem is well defined and the choice of GNN architecture is well motivated.
+ The result regarding the model handling the repetitive regions more effectively than Raven is particularly important for the use of the proposed method for de novo genome assembly.

Weaknesses:
- The results in the paper are not comprehensive enough to support and highlight the claims in the introduction. They are restricted to a very small subset of the simulated datasets generated from the human genome (focusing on just 3 chromosomes).
- The choice of the baselines is not entirely clear. While Raven seems like a reasonable assembly method, the results in the paper (Vaser et al, 2021) do not suggest that it is the best assembly tool, thus requiring at least a mention/discussion of other genome assembly methods in the literature.
- Furthermore, the baseline Raven paper explores different genomes (plants, prokaryotes, eukaryotes) as well as different sequencing technologies. This paper would benefit from a similarly comprehensive study of the performance of the GNN based method across different datasets
- The GNN method requires the ground-truth position information that is obtained from an assembled genome and reads generated via simulation. Is the inference of a model trained from simulations then generalizable to another chromosome or another genome with actual reads? The results seem to suggest the former but it is unclear if the inferences were performed on actual reads of other chromosomes. How will such a framework work for de novo assembly?
- Does the execution time include the training time of the GNN model? If not, then are the execution time comparisons fair?
- The scalability of the method should be further explored in comparison to other baselines as graphs with the execution time as a function of the size of the mini references scaling to sizes>10 Mbp.
- It would be useful to explore the potential cause of the differences in alignment performance of the GNN method (and others) across 2Mbp, 5Mbp, and 10Mbp mini-references.


**Summary Of The Paper:**

This paper presents an interesting idea of using graph-based neural networks (GNNs) to find an optimal layout path during de novo genome assembly using the Overal-Layout-Consensus (OLS) method. Existing methods perform this step by using heuristics to simplify the graph (removing edges/nodes)  instead of finding a path to assemble the genome from the assembly graph. Therefore, they fail to resolve complex graphs with repetitive regions and instead remove or trim those altogether. The paper proposes to train a gated GNN to predict probabilities for the edges of an assembly graph. These probabilities can help in defining a path on the assembly graph that can lead to faster de novo assembly during the Layout phase. To obtain ground truth positional information, the paper uses simulated reads generated from the human genome. The GNN model is trained on mini-references (2 Mbp regions) taken from chromosome 11 and the alignment and execution time results are presented for 2Mbp, 5Mbp, and 10Mbp mini-references from chromosomes 11,10, and 12. The results show that the proposed GNN strategy improves over the greedy approach in alignment performance and an existing assembler (Raven) in execution time.


**Summary Of The Review:**

The proposed idea in the paper is interesting, however, the limited set of results are insufficient to support its claims and highlight its usefulness for de novo genome assembly.

---

> ### Author Response · Authors · 2021-11-18
> **Response to Reviewer xJLL**
>
> We thank the reviewer for the insightful review.
>
> Due to some frequently received comments, we decided to address them as a general reply found here: https://openreview.net/forum?id=1QxveKM654&noteId=ZnkbPVBdr_7
>
>
> - Small subset of the simulated datasets.
>
> We agree with the reviewer that dataset was rather small. Our intention was to show the proof of concept and to the best knowledge this the first approach for replacing the whole layout phase with a method based on GNNs.
>
> - Choice of baseline
>
> Our goal was to show that the Layout phase in Raven can be replaced with GNNs instead of heuristics and that this could bring benefits in terms of contiguity and speed. The data on which the GNN was trained and evaluated was generated by the Overlap phase of Raven, and since this step varies greatly between the assemblers, we did not involve them into the evaluation. Some assemblers use completely different paradigm (i.e. de Bruijn graph assemblers) and other use a different strategy for Overlap phase and these strategies might have influence on the Layout phase. Furthermore, most of assemblers do not output results of the layout phase and we would need for each of them to learn the source code which would take a lot of time. We chose Raven because we are familiar with its source code.
>
> Raven is one of the state of the art assemblers and our intention was to prove that using GCN approach for graph simplification might be a good alternative. We deem that this opens new avenues for other researchers who might try the similar approach for other assemblers. In our future work we plan to replace Raven’s layout phase with a GNN, and then we will compare the final product with other state-of-the-art assemblers.
>
>
> - Different genomes
>
> It is true that this paper would benefit from evaluation on different genomes, but for now we focused on human data since for this we have a high quality reference and it's easy to evaluate the performance this way.
>
>
> - Generalization
>
> We have not yet tested on actual reads of other chromosomes/genomes, so we cannot make claims that it is generalizable. This is our next step.
>
>
> - Execution time
>
> The execution time does not include the training time of the GNN model. We believe this is a fair comparison, since the training time is something that was done once and we would not expect that users train the model themselves. Once trained, the model could be used on other datasets.
>
>
> - Scalability
>
> We agree that we need to further need to assess this approach using various genome sizes

---

> > ### Comment · Reviewer_xJLL · 2021-11-28
> > **Response to the rebuttal**
> >
> > Thank you for your response to my comments. I completely agree that this paper is presenting a very interesting idea of using GNNs for the Layout phase of the assembly. However, the paper in its current form requires further investigation and experiments to make the work truly impactful and useful for the community. Therefore, I would like to retain my original score.

---

### Official Review · Reviewer_GFZQ · 2021-11-02

**Correctness:** 3
**Technical Novelty And Significance:** 4
**Empirical Novelty And Significance:** 2
**Recommendation:** 5
**Confidence:** 2

**Main Review:**

Strengths
- uses synthetic data to allow for quantifying accuracy of reconstructed genomes
- benchmark against a recent previous method, Raven, was performed for both accuracy and execution time

Limitations:
- The synthetic sequencing reads do not contain any errors. This may  be a reasonable approximation only for HiFi reads. The authors should include simulations with varying levels of noise in order to test the efficacy of the proposed method versus Raven. If this is not done, then the scope of the paper should be about genome assembly for HiFi reads, instead of the more generic language used throughout the text.
- How does the method do with shorter reads? Perhaps a sweep of this could be informative to set lower bounds. This is important for practitioners who may erroneously use this thinking it is a general assembler.
- The range of synthetic genomes are very small, up to 10 Mbp. As the authors stated, a human chromosome is 250 Mbp and there are of course 23 pairs of chromosomes.  Thus, it is unclear how well the GCN can assemble realistic data (not HiFi) for which there is plenty of data. A good use case could be metagenomics, where de novo assembly of contigs is critical. There needs to be some demonstration on real data. A proof of principle using real data would make this paper more compelling.
- With Raven, there is no training. It assembles the genome with an algorithm. On the other hand, the GNN has parameters that require training. So, the reported execution time for GNN includes just the inference time and not the training, right? Is it fair to compare just the inference time of the GNN on a very powerful A100 vs the whole end-to-end task for the other methods? Perhaps an end-to-end execution time is more appropriate. Also, what is the memory requirement of the GCN? Do people need 48GB of GPU ram? How would the model's execution time be on a GPU that is more common, like a P100 on google colab?
- Can this method deal with allelic differences across chromosomes?
- The execution time was based on a single Intel Xeon E5-2698 v4 CPU, which has 20 cores. The Raven paper states that they were run on two AMD EPYCTM 7702 64-core processors. How much would this speed up the execution time, using more cores for Raven.

**Summary Of The Paper:**

This paper introduces a graph convolutional network to assemble a genome from long, perfectly accurate sequencing reads. They show that their model performs better when considering overlap lengths and graph topology versus just the overlap lengths. The proposed method is compared to other methods, such as Raven, and shown to reconstruct the genome much faster. The work was demonstrated on synthetic data without any read errors and thus serves as a prototype for real de novo genome assembly problems, which are much longer and contain noise.

**Summary Of The Review:**

This paper provides a promising approach to use GCNs to assemble genomes. To the best of my knowledge, this is novel. The demonstration is focused on synthetic data with no noise. While this can be reasonable approximation with HiFi reads, it remains unclear how it extends to other kinds of data, eg. short- and long-read RNA seq for which there exists many data. To be impactful, a comparison on more realistic synthetic data is important, such as noise added and longer genomes. While there are many good things about this paper to be excited by, critically, a demonstration on real data is needed.

---

> ### Author Response · Authors · 2021-11-18
> **Response to Reviewer GFZQ**
>
> We thank the reviewer for the insightful review.
>
> Due to some frequently received comments, we decided to address them as a general reply found here: https://openreview.net/forum?id=1QxveKM654&noteId=ZnkbPVBdr_7
>
>
> - Reads do not contain any errors
>
> We completely agree with the first point, and acknowledge oversight on our side. The main goal was to focus on HiFi data, not other types, and the more specific language would be appropriate.
> However, we argue that once graph is constructed there is little influence of read errors on graph simplification. We focused our research on the layout phase when the graph is constructed and our aim was to prove that in this situation GNN approach might replace existing algorithms.
>
> - Shorter reads
>
> The main problem in genome assembly are repetitive regions which might be very long. The only possible solution is to find extra long reads which can bridge them or use tiny differences in repetitive segment which require both long and high-quality reads. Therefore, long reads are de facto standard at the moment in the field. Between very long Oxford Nanopore reads and long and high fidelity Pacbio reads we chose the latter because recently it has been shown that assemblies with Pacbio Hifi reads outperform assemblies with Nanopore reads.
>
> We apologize that we do not clarify that in the paper and stated that we are working towards building a "high-quality long reads assembler"
>
> - Real data
>
> Metagenome assembly was not the scope of this work because there are differences between single genome and metagenome assemblers. The aim and problems are slightly different. However, we agree that showing the performance on larger data, such as whole chromosomes, is important and that generalization from 2 Mbp to 10 Mbp does not imply generalization to 250 Mbp as well.
>
> - No training for Raven
>
> That is correct, the reported time is only for the inference. However, once trained, the model could be used for other genome assembly tasks and there would be no need to train the model again. Therefore, we argue there is no need to include the training time into the evaluation.
>
> - Allelic differences across chromosomes.
>
> No. At the moment we would like to focus on a simpler task – haploid genome assembler. Notice also that (a) our data is generated and thus there is no alleles in it, (b) the CHM13 is a haploid genome, meaning even if we had used real CHM13 data, there would still be just one allele per gene.
>
>
> - Execution time
>
> In our case, almost not at all. In the evaluation of the execution time, we did not include the time needed to generate the graph, since that time is the same for both Raven and our method (Overlap phase). Our model can be viewed as a substitution for the Layout phase of Raven. Most of the Layout phase is single-thread algorithms, so the evaluation was single-thread Layout in Raven vs single-thread forward pass in GNN.
> The only part of the Layout phase in Raven which utilizes multi-threading is the Force-Directed Layout algorithm, which was in our case was not too time consuming even on a single thread due to relatively small graphs. With larger graphs, the difference between single-threading and multi-threading could become more prominent.

---

### Official Review · Reviewer_Av24 · 2021-11-03

**Correctness:** 3
**Technical Novelty And Significance:** 2
**Empirical Novelty And Significance:** 2
**Recommendation:** 5
**Confidence:** 5

**Main Review:**

Existing de novo assembly algorithms or software require extreme computational resources in both time and space. Neural network-based de novo assembly can reduce these extreme costs; therefore, this paper has nontrivial contributions on genome assembly area, and I think this work is promising.

On the other hand, it would be great if the authors address issues as follows:

●	Related work

I think this paper is one of the applications or the extensions of existing work [1], [2], [3], [4]. Could the authors briefly introduce those topics or research areas in the revised manuscript?

●	Experiment (dataset)

In this work, the authors used the human reference genome (CHM13) to demonstrate that the model has a capacity to reconstruct the reference sequence. However, as reference genomes such as CHM13 and GRCh38 already exist, biological researchers and practitioners do not need to assemble the human genome at usual times. Thus, conducting the experiment to explore the generalizability or transferability on unseen (i.e., non-human) sequences is essential to further demonstrate the utility of the proposed method. In other words, the proposed method should work well on assembly graphs of other species, which do not have reference genomes.

●	Experiment (evaluation)

To measure the accuracy of the assembly, the authors compared the lengths of the reconstructed genome. Obviously, this cannot exactly represent the accuracy of the assembly. Thus, I think the authors should supplement other accuracy metrics in the revised version.

[1] Veličković et al., “Neural Execution of Graph Algorithms”, ICLR 2020.

[2] Vrček et al., “A step towards neural genome assembly”, NeurIPS 2020 workshop.

[3] Joshi et al., “Learning TSP Requires Rethinking Generalization”, CP 2021.

[4] Cappart et al., “Combinatorial Optimization and Reasoning with Graph Neural Networks”, IJCAI 2021.


**Summary Of The Paper:**

This paper addresses the problem of de novo assembly, which is one of the most complicated tasks in bioinformatics. De novo assembly is a task to reconstruct the whole genomic sequence given numerous broken pieces of the original sequence. Existing de novo assembly approaches, based on conventional graph algorithms (e.g., de bruijn graph and string graph) require extreme computational cost, restricting further applications.

In this paper, the authors proposed a novel method that reconstructs the original sequence from the assembly graph by using a graph neural network (GNN). When the assembly graph is fed to the GNN, the model outputs the paths to reconstruct the original sequence. The authors demonstrated the effectiveness of the proposed method by comparing reconstruction time and accuracy (i.e., length of the reconstructed genome) with one of the de novo assembly software (Raven) on human genome reference data.


**Summary Of The Review:**

This paper proposed a novel neural network-based de novo assembly method. Since the proposed method can reduce extreme computational costs during the genome reconstruction process, it could be a promising direction for de novo assembly algorithms.

However, the utility of the proposed method is not sufficiently demonstrated from the perspective of biology researchers and practitioners. Also, the technical novelty of the proposed method seems to be limited in terms of machine learning techniques.

Therefore, I am leaning towards rejecting this paper at this moment.

---

> ### Author Response · Authors · 2021-11-18
> **Response to Reviewer Av24**
>
> We thank the reviewer for the insightful review.
>
> Due to some frequently received comments, we decided to address them as a general reply found here:
> https://openreview.net/forum?id=1QxveKM654&noteId=ZnkbPVBdr_7
>
>
> - Related work
>
> We agree, a section about related work would certainly be useful, and we will make sure to include it.
>
>
> - Experiment (dataset)
>
> The reviewer claims that, due to existence of human reference genomes such as CHM13 and GRCh38, the researchers would not need to assemble human genomes once again. We used CHM13 for our evaluation because it is the first telomere to telomere reconstructed large genome. For resolving this genome authors manually inspected large parts of the graph in the layout phase. Our intention was to show that using graph neural networks is a promising approach to replace both existing algorithms for layout phase and manual curation.
>
> At the moment there is a large effort of scientific community to construct a human pangenome which would involve assembled genomes from various ethnicities. Structural variation between ethnicities might be significant and are easier to capture by de novo genome assembly than by using the reference genome. Therefore, even on human data, de novo genome assembly is still not completely solved.
>
> Finally, we do agree that testing the generalizability to non-human genomes is crucial for having a general-purpose assembly and we will include that in the future work.
>
>
> - Experiment (evaluation)
>
> We completely agree with the reviewer that the length of the reconstructed genome is not suitable for evaluation of the assembly genomes in general.
>
> However, our intention in this paper was not producing a new de novo assembler. De novo assembly is a complex process with many phases. Our focus was on the layout phase and we intended to show that on this task our method can compete with both greedy approach and existing implementation of layout phase starting from the same initial graph.

---

> > ### Comment · Reviewer_Av24 · 2021-11-29
> > **Comments for the rebuttal**
> >
> > Thank the authors for the response, and I agree that the proposed idea is promising in genome assembly. However, although my concern about human genome reference has been resolved, I think the current submission should supplement experiments to show the effectiveness or impact of the proposed idea. Therefore, I would like to keep my initial evaluation.

---

### Author Response · Authors · 2021-11-18
**General Response and Clarification**

We sincerely thank all the reviewers for their valuable comments. Since most of the comments were related to de novo assembly perspective of the paper, we understand that we needed to describe our aim clearer.

Our goal was not at this phase to build a new tool for de novo assembly. We aimed to prove that one phase of de novo assembly, Layout phase (graph simplification), can be replaced with methods based on graph neural networks. To create a graph, we used overlap phase of Raven assembler. Because of this we aimed to show that we can achieve similar results in comparison with Raven on the same graph, using a deep learning approach. Raven uses OLC paradigm and among OLC based assemblers there are no big differences in layout phase (removal of transitive edges and tips, bubble popping,…) and we hope that this might motivate other researchers to try to use GNNs for layout phase.

---

### Decision · Program_Chairs · 2022-01-20

**Decision:**

Reject

**Comment:**

The paper demonstrates that one phase of de novo assembly, specifically the layout phase, can be replaced with graph-neural-network based methods. The paper clarifies in the rebuttal that it focuses on building a method for assembling high-quality long reads.

All four reviewers rated the paper as below the acceptance threshold. The reviewers largely agree that the idea of using GNNs to assemble a genome from reads is novel, interesting, and has the potential to be very useful.
The reviewers raise the following concerns: The paper only considers synthetic data, and the synthetic reads used in the simulations are error-free. In practice, reads are not error-free, and thus simulations on real data or at the very least on reads with errors are needed. The authors acknowledge that, and state that they'll provide such experiments in future work. In summary, the reviewers found the experiments to be insufficient to support the claims, even though it is understood by the reviewers and me that the paper only presents a proof-of-concept idea. I agree with the reviewers that simulations on erroneous reads, ideally real data, would be needed for acceptance.

I recommend to reject the paper, since the paper provides insufficient experiments to understand the merits of the proposed approach.